# Explainable artificial intelligence model to predict acute critical illness from electronic health records

Simon Meyer Lauritsen [1,2 ✉], Mads Kristensen[1], Mathias Vassard Olsen[3], Morten Skaarup Larsen[3], Katrine Meyer Lauritsen [2,4], Marianne Johansson Jørgensen[5], Jeppe Lange[2,5] & Bo Thiesson[1,6]

Acute critical illness is often preceded by deterioration of routinely measured clinical parameters, e.g., blood pressure and heart rate. Early clinical prediction is typically based on manually calculated screening metrics that simply weigh these parameters, such as early warning scores (EWS). The predictive performance of EWSs yields a tradeoff between sensitivity and specificity that can lead to negative outcomes for the patient. Previous work on electronic health records (EHR) trained artificial intelligence (AI) systems offers promising results with high levels of predictive performance in relation to the early, real-time prediction of acute critical illness. However, without insight into the complex decisions by such system, clinical translation is hindered. Here, we present an explainable AI early warning score (xAI-EWS) system for early detection of acute critical illness. xAI-EWS potentiates clinical translation by accompanying a prediction with information on the EHR data explaining it.

[1] Enversion A/S, Fiskerivej 12, 1st floor, 8000 Aarhus C, Denmark. [2] Department of Clinical Medicine, Aarhus University, Palle Juul-Jensens Boulevard 82, 8200 Aarhus N, Denmark. [3] Department of Biomedical Engineering and Informatics, Aalborg University, Niels Jernes Vej 12, 9220 Aalborg Ø, Denmark. [4] Department of Endocrinology and Internal Medicine, Aarhus University Hospital, Palle Juul-Jensens Boulevard, 99, 8200 Aarhus N, Denmark. [5] Department of Research, Horsens Regional Hospital, Sundvej 30, 8700 Horsens, Denmark. [6] Department of Engineering, Aarhus University, Inge Lehmanns Gade 10, 8000 Aarhus C, Denmark. ✉email: sla@enversion.dk

Artificial Intelligence (AI) is capable of predicting acute critical illness earlier and with greater accuracy than traditional early warning score (EWS) systems, such as modified EWSs (MEWSs) and sequential organ failure assessment scores (SOFAs)[1–13]. Unfortunately, standard deep learning (DL) that comprise available AI models are black-box predictions that cannot readily be explained to clinicians. A tradeoff must, therefore, be made between transparency and predictive power, which for high-stake applications most often favor the simpler, more transparent systems, where a clinician can easily back-trace a prediction. To benefit from the higher predictive power, the importance of explainable and transparent DL algorithms in clinical medicine is without question, as was recently highlighted in the Nature Medicine review by Topol[14].

Transparency and explainability are an absolute necessity for the widespread introduction of AI models into clinical practice, because an incorrect prediction may have grave consequences[15–18]. Clinicians must be able to understand the underlying reasoning of AI models so they can trust the predictions and be able to identify individual cases in which an AI model potentially gives incorrect predictions[15–19]. Consequently, a useful explanation involves both the ability to account for the relevant parts in an AI model leading to a prediction, but also the ability to present this relevance in a way that supports the clinicians causal understanding in a comprehendible way[20]. An explanation that is too hard to perceive and comprehend will most likely not have any practical effect.

In this work, we will present explainable AI early warning score (xAI-EWS), which comprises a robust and accurate AI model for predicting acute critical illness from electronic health records (EHRs). Importantly, xAI-EWS was designed to provide simple visual explanations for the given predictions. To demonstrate the general clinical relevance of the xAI-EWS, we present results here from three emergency medicine cases: sepsis, acute kidney injury (AKI), and acute lung injury (ALI). The xAI-EWS is composed of a temporal convolutional network (TCN)[21,22] prediction module and a deep Taylor decomposition (DTD)[23–27] explanation module, tailored to temporal explanations (see Fig. 1).

The architecture of the TCN has proven to be particularly effective at predicting events that have a temporal component, such as the development of critical illness[5,22,28,29]. The TCN operates sequentially over individual EHRs and outputs predictions in the range of 0–100%, where the predicted risk should be higher for those patients at risk of later acute critical illness, compared to those who are not. The DTD explanation module delineates the TCN predictions in terms of input variables by producing a decomposition of the TCN output on the input variables[30,31].

## Results

**Predictive performance**. In Fig. 2, the predictive power of the xAI-EWS is presented in summary form with results from the onset time to 24 h before onset. Area under the receiver operating characteristic curve (AUROC) with mean values and 95% confidence intervals (CIs) over the five cross-validations folds were 0.92 (0.9–0.95)–0.8 (0.78–83), 0.88 (0.86–0.9)–0.79 (0.78–0.8), and 0.90 (0.89–0.92)–0.84 (0.82–0.85) for sepsis, AKI, and ALI, respectively. Area under the precision-recall curve (AUPRC) with mean values and 95% CIs were 0.43 (0.36–0.51)–0.08 (0.07–0.09), 0.22 (0.19–0.24)−0.13 (0.12–0.14), and 0.23 (0.21–0.26)–0.23 (0.22–0.24) for sepsis, AKI, and ALI, respectively. (Supplementary Tables 1 and 2).

**Explanations**. The xAI-EWS enabled two perspectives on the model explanations: an individual and a population-based perspective. For the individual perspective, the explanation module enabled the xAI-EWS to pinpoint which clinical parameters at a given point in time were relevant for a given prediction. In current clinical practice, the workflow normally follows that clinicians observe either a high EWS or an increase in EWS. However, the following targeted clinical intervention concerning the potential critical illness happens when the clinician understands which clinical parameters have caused the high EWS or the change in EWS. This is one of the main reasons why AI-based EWS systems need to be able to explain their predictions. The xAI-EWS system we developed allows for such explanations in real time and across all clinical parameters used in the model. An example of an output from the explanation module, utilizing the individual perspective, is illustrated in Fig. 3. Individual clinical parameters are sized according to the amount of back-propagated relevance. Figure 3a shows the 10 most relevant parameters with respect to sepsis for a single patient with a risk score of 76.2%. High respiration frequency, high pulse rate, and low plasma albumin are the most important predictors of sepsis. The

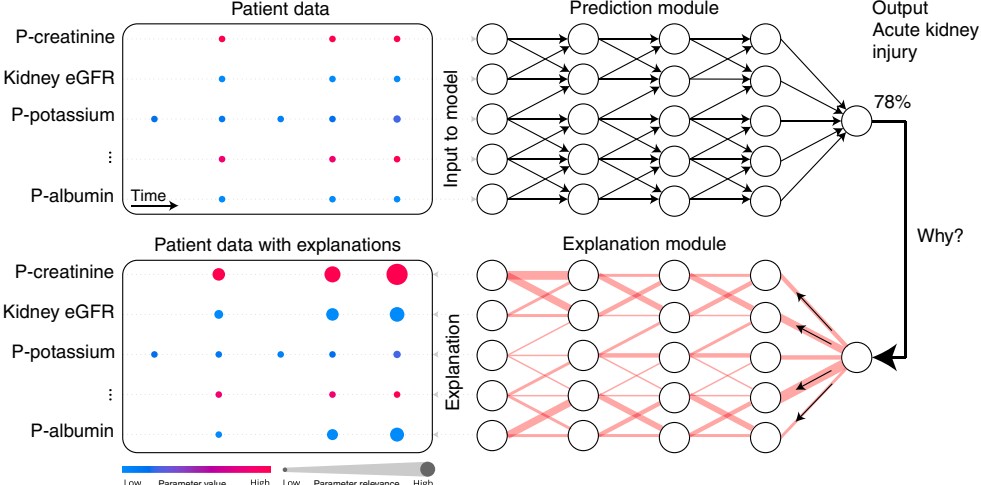

**Fig. 1 Overview of the xAI-EWS system.** Each patient's data from the EHR is used as input in the TCN prediction module. Based on this data, the model makes a prediction, such as a 78% risk of AKI. The DTD explanation module then explains the TCN predictions in terms of input variables. P, plasma; eGFR, estimated Glomerular filtration rate; DTD, deep Taylor decomposition; TCN, temporal convolutional network; xAI-EWS, explainable artificial intelligence early warning system.

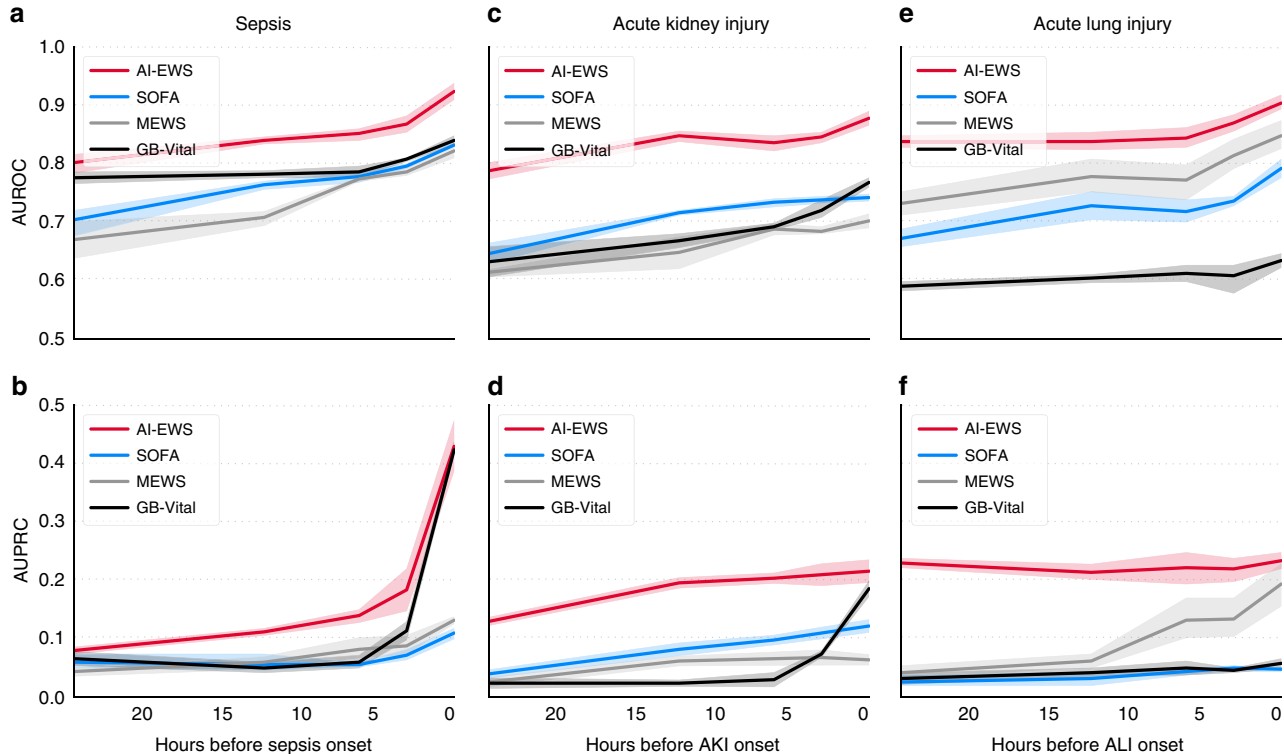

**Fig. 2 Predictive performance of the xAI-EWS.** The xAI-EWS results are compared with those from MEWS, SOFA, and the gradient boosting vital sign model (GB-Vital). Predictive performance is shown from the onset time to 24 h before onset. AUROC performance is shown for sepsis (**a**), AKI (**c**), and ALI (**e**), and AUPRC performance is shown for sepsis (**b**), AKI (**d**), and ALI (**f**). The solid lines indicate mean values. Lighter semitransparent colors surrounding the solid lines indicate uncertainty by 95% confidence intervals calculated from the five test datasets ($n = 163,050$ patients examined over 5 cross-validation folds with a test size of 10%).

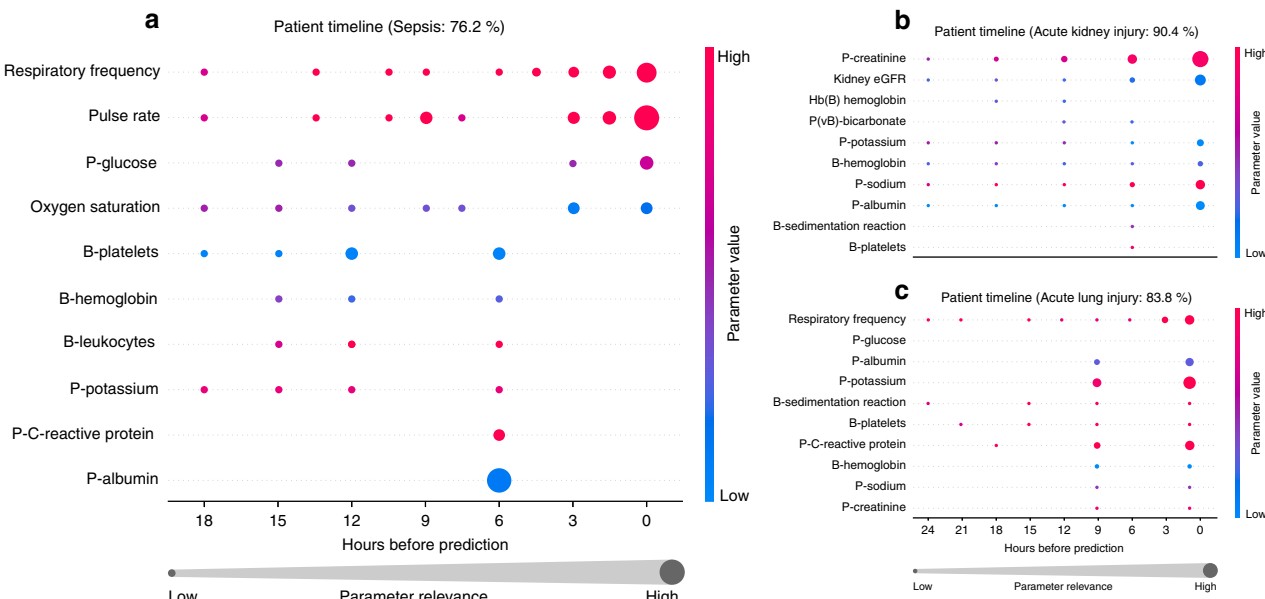

**Fig. 3 Results from the explanation module displays for three individual patients.** Three selected patient timelines with back-propagated relevance for sepsis (**a**), AKI (**b**), and ALI (**c**) are shown. Only the 10 highest-ranking parameters in descending order by the mean relevance are displayed. The data shown in the three timelines match the data from the observation window, such that a time equal to zero is the prediction time. The data-points are colored according to the 5th and 95th percentiles for each parameter across the whole dataset. The blue data-points correspond to a value between 0 and the 5th percentile, the red data-points correspond to a value between the 95th and 100th percentiles, and data-points with values close to the median are purple. P, plasma; eGFR, estimated Glomerular filtration rate.

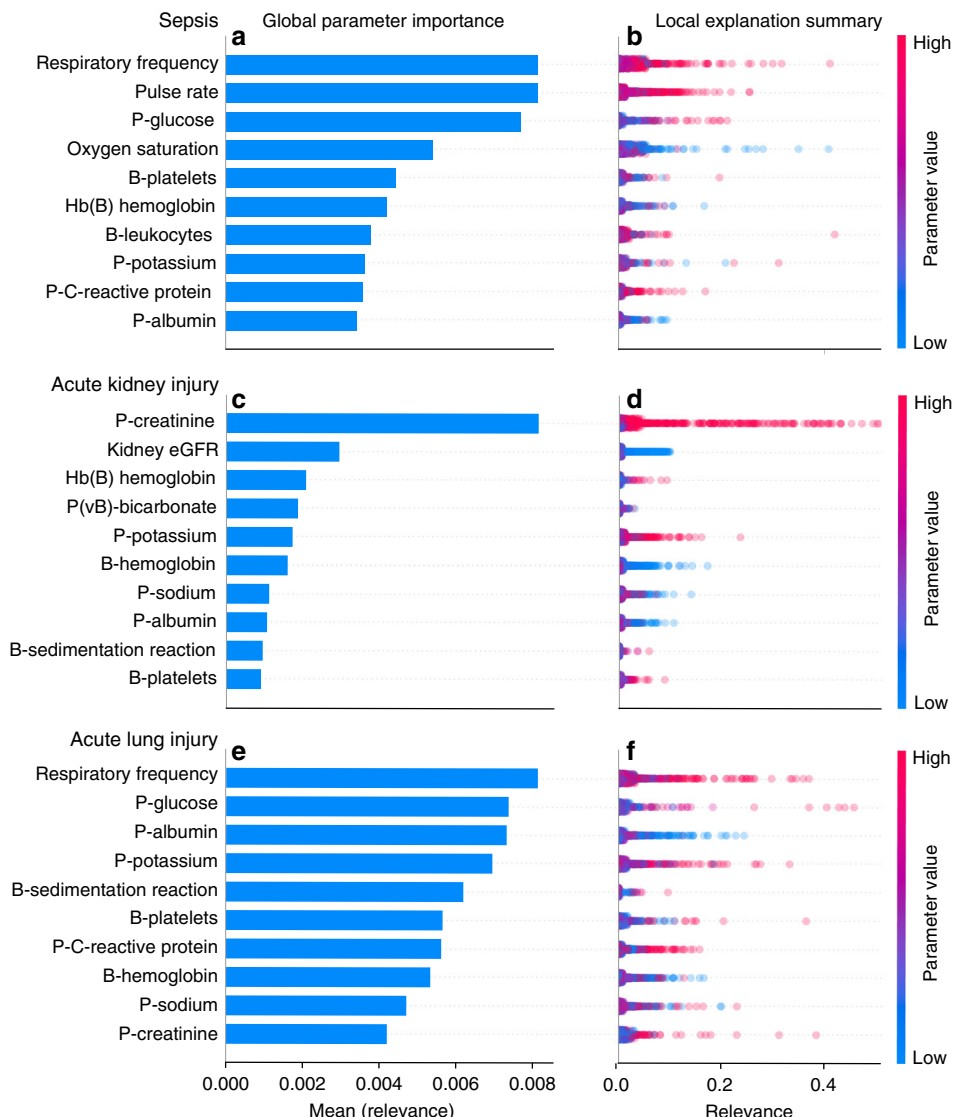

**Fig. 4 Results from the explanation module displaying the global parameter importance and local explanation summary.** The parameters are sorted in descending order according to global parameter importance, defined by the mean relevance, which is identified by the blue horizontal bars for sepsis (**a**), AKI (**c**), and ALI (**e**). The local explanations summary shows all the individual data-points, colored by parameter value and displaced by the mean relevance for sepsis (**b**), AKI (**d**), and ALI (**f**). The height of the data-points shown for each parameter in the local explanation summary correlate with the number of data-points at the associated level of relevance. The population-based perspective is simplified by ignoring the temporal relevance variations, treating all data-points at different times equally. P, plasma; eGFR, estimated Glomerular filtration rate.

physiological values of respiration frequency and pulse rate do not seem to increase close to the prediction time, but, inspecting the increasing sizes of relevance, it appears that the xAI-EWS attributes more weight to recent values. Figure 3b and c show the 10 most relevant parameters with respect to AKI and ALI for two patients with risk scores of 90.4% and 83.8%, respectively.

In terms of the population-based perspective, the xAI-EWS is able to facilitate transparency and, thereby, induce trust, by giving clinicians insights into the internal mechanics of the model without any deep technical knowledge of the mechanisms behind it.

In Fig. 4, the 10 most important clinical parameters for each of the three models are shown. The parameters are sorted by the decreasing mean relevance as computed for the local, back-propagated relevance scores across the entire population, but only for patients who were positive for sepsis, AKI, or ALI.

The blue horizontal bars in the left column of Fig. 4 display the mean relevance. In the local explanation summary in the right-hand column of Fig. 4, the distribution of the back-propagated relevance scores for each clinical parameter are shown and color-coded by the parameter value associated with the local explanation. As an example, in Fig. 4c and d, the AKI model seems to associate high P-creatinine levels and low estimated glomerular filtration rates with AKI. When the model is confident about a decision, it will output a high probability. This high probability will result in more relevance available for distributing backward; it will also result in larger relevance scores. On the contrary, when the model does not believe that a patient will develop an acute critical illness, it will output a low probability, and the associated relevance scores will also be low. The summary distribution allows clinicians to get an idea of what to expect from the model in clinical practice.

## Discussion
In this study, we present xAI-EWS—an explainable AI early warning score system for early detection of acute critical illness.

While maintaining a high predictive performance, our system explains to the clinician on which relevant EHRs data the prediction is grounded.

Previous work has employed different strategies to develop explainable prediction models[13,32–36]. RNN variations with attention have been suggested for illness severity assesment[13], risk of hospitalization prediction[32], sepsis prediction, and myocardial infarction prediction[35]. Shickel et al.[13] developed an interpretable deep learning framework called DeepSOFA that leveraged temporal measurements to assess illness severity at any point during an ICU stay. An RNN with gated recurrent units (GRU) and self-attention was proposed to highlight particular time steps of the input time series that the model believed to be most important in formulating its mortality prediction. Kaji et al. demonstrated how attention can be applied at the level of input variables themselves when predicting outcomes for ICU patients[35]. Choi et al. used a factorized approach to compute attention over both variables and time using embedded features rather than the immediate input features themselves[36]. Zhang et al. compressed the entire patient EHR into a complete vector representation and used GRU and self-attention to predict the future risk of hospitalization in an interpretable framework called Patient2Vec[32]. Our work differentiates from the above studies by utilizing TCN and LRP instead of RNNs with attention.

It is important to note that the xAI-EWS presented in this study should not be conceived as the one-and-only multi-outcome model. Rather, it should be viewed as a general method of building precise and explainable models for acute critical illness. Following this line of thinking, it is obvious that other models with important critical outcomes, such as hypokalemia, hyperkalemia, acute constipation, and cardiac arrest, should be added to the three models presented in this study. This will result in a series of EWS models that are all specialists in their respective fields.

One important point to note is that more work is needed to investigate better ground truth definitions of the evaluated critical illnesses, such as AKI and ALI. We based the ground truth on the need for continuous positive airway pressure (CPAP) or non-invasive ventilation (NIV) because $PaO_2/FiO_2$ measurements were not available. The KDIGO is an indicator of AKI that has a long lag time after the initial renal impairment, as mentioned by Tomašev et al.[1]. Our model is trained and has been tested on a large dataset that is highly representative of the Danish population. However, validating the predictive performance of the xAI-EWS on a different population would make for an interesting study, and, as the xAI-EWS currently uses just 33 clinical parameters, this appears feasible. An interesting subject for further study would also be to compare how well the explanation module in this paper conveys explanations to the clinical experts in various contexts compared to alternatives. To that end, Holzinger et al.[37] recently proposed a Likert-scale based method tailored to explanations from AI.

We limited the length of the observation window to 24 h to ensure that the model was based on clinical, and time-relevant, features. A variable-length window greater than 24 h should be explored in an upcoming study.

Model development was done in an iterative way where results from technical development were continuously discussed with clinicians from an emergency department. The purpose of this process was to ensure that the models learned at least some correlations that are already considered established knowledge in the clinical field. It would be obvious to try to use this technology hypothesis-generating, whereby output from LRP analysis is used as inspiration to discover new and unknown correlations.

The low prevalence of the sepsis, AKI, and ALI (2.44%, 0.75%, and 1.68%) resulted in a very unbalanced classification problem.

**Table 1 Patient population description.**

| | |
|---|---|
| Unique patients, no. | 66,288 |
| Unique admissions, no. | 163,050 |
| Age, median years | 55.2 |
| Gender, male, % of total admissions | 45.9 |
| Length of stays, median hours | 153.6 |
| Laboratory measurements, average per admission | 39 |
| Hospital mortality, % of total admissions | 0.85 |
| Sepsis present, % of total admissions | 2.44 |
| Acute Kidney Injury present, % of total admissions | 0.75 |
| Acute Lung Injury present, % of total admissions | 1.68 |

To combat this imbalance, we tried to oversample the positive class with replacements. The oversampling did not affect model performance, but stretched the output probabilities into a wider range. The results reported were computed without resampling.

In summary, we have presented the xAI-EWS—an explainable AI EWS system for the prediction of acute critical illness using EHRs. The xAI-EWS shows a high predictive performance while enabling the possibility to explain the predictions in terms of pinpointing decisive input data to empower clinicians to understand the underlying reasoning of the predictions. We hope that our results will be a steppingstone toward a more widespread adoption of AI in clinical practice. As stated, explainable predictions facilitate trust and transparency—properties that also make it possible to comply with the regulations of the European Union General Data Protection Regulation, the Conformité Européenne (CE) marking, and the United States Food and Drug Administration[38].

## Methods

**Data description**. In this study, we analyzed the secondary healthcare data of all residents of four Danish municipalities (Odder, Hedensted, Skanderborg, and Horsens) who were 18 years of age or older for the period of 2012–2017. The data contained information from the electronic health record (EHR), including biochemistry, medicine, microbiology, and procedure codes, and was extracted from the "CROSS-TRACKS" cohort, which embraces a mixed rural and urban multi-center population with four regional hospitals and one larger university hospital. Each hospital comprises multiple departmental units, such as emergency medicine, intensive care, and thoracic surgery. We included all 163,050 available inpatient admissions (45.9% male) during the study period and excluded only outpatient admissions. The included admissions were distributed across 66,288 unique residents. The prevalence for sepsis, AKI, and ALI among these admissions was 2.44%, 0.75%, and 1.68%, respectively (see Table 1). The CROSS-TRACKS cohort offers a combined dimensional model of the secondary healthcare data. Merging all data sets is possible via a unique personal identification number given to all Danish citizens and by which all information within any public institution is collected[31].

The model parameters were limited to include 27 laboratory parameters and six vital signs (see Tables 2 and 3). The parameters were selected by trained specialists in emergency medicine (medical doctors) with the sole purpose of simplifying the model to enable a better discussion of the model explanations. While a deeper model with more parameters might lead to better performance, it would also have made the discussions between clinicians and software engineers difficult. Therefore, the scope of this article is not to obtain the best performance at all costs but to demonstrate how clinical tasks can be supported by a fully explainable deep learning approach.

**Data preprocessing**. In the data extracted from the CROSS-TRACKS cohort, each admission is represented as a time-ordered sequence of EHR events. Importantly, the time-stamped order of this data reflects the point in time at which the clinicians record each event during the admission. Each event comprises three elements: a time stamp; an event name, such as blood pressure; and a numerical value. The event sequence is partitioned in aggregated intervals of one hour, such that the observation window of 24 h is divided into 24 one-hour periods, and all the events occurring within the same one-hour period are grouped together by their average numerical value.

**Gold standards**. Via a classification process, each admission was classified as sepsis-positive, AKI-positive, ALI-positive, or negative (no critical illness). For sepsis classification, we followed the recent Sepsis-3[30,39] implementation by Moor et al.[5], according to which both suspected infection and organ dysfunction are

**Table 2 List of clinical parameters.**

| Laboratory parameters | | |
|---|---|---|
| P(aB)-Hydrogen carbonate | P(aB)-Potassium | P-Creatinine |
| P(vB)-Hydrogen carbonate | B-Hemoglobin | P-Bilirubin |
| P(aB)-pO$_2$ | B-Neutrophils | P-Prolactin |
| P(vB)-pCO$_2$ | B-Eythrocyte sedimentation rate | P-Glucose |
| P(aB)-pCO$_2$ | B-Platelets | P-C-reactive protein (CRP) |
| P(aB)-pH | B-Leukocytes | Hb(B)-Hemoglobin A1c |
| P(vB)-pH | P-Sodium | Glomerular filtration rate (eGFR) |
| P(aB)-Lactate | P-Potassium | |
| P(aB)-Sodium | P-Lactate dehydrogenase (LDH) | |
| P(aB)-Chloride | P-Albumin | |
| *Vital sign parameters* | | |
| Systolic blood pressure | Respiratory frequency | SpO$_2$ (Pulsoxymetry) |
| Diastolic blood pressure | Pulse | Temperature |

required to be present[5,30,39]. Suspected infection was defined by the co-occurrence of body fluid sampling and antibiotic administration. When a culture sample was obtained before antibiotics administration, the antibiotic had to be ordered within 72 h. If the antibiotic was administered first, then the culture sample had to follow within 24 h.

The degree of organ dysfunction is described by an acute increase in the SOFA[40] score and an increase of more than or equal to two points is used in the criteria for sepsis[39]. To implement the organ dysfunction criterion, we used a 72-h window from 48 h before to 24 h after the time of suspected infection, as suggested by Singer et al.[30] and Moor et al.[5]. The Sepsis-3 implementation is visualized in Fig. 5.

AKI classification was performed according to the KDIGO criteria[31]. KDIGO accepts three definitions of AKI: (1) an increase in serum creatinine of 0.3 mg/dl (26.5 μmol/l) within 48 h; (2) an increase in serum creatinine by 1.5 times the habitual creatinine level of a patient within the previous seven days; and (3) a urine output of <0.5 ml/kg/h over 6 h. Following the work of Tomašev et al.[1], only the first two definitions were used to provide ground-truth labels for the onset of AKI as urine measurements were not available. The habitual creatinine level was computed as the mean creatinine level during the previous 365 days. We used binary encoding for AKI such that all three severity stages (KDIGO stages 1, 2, and 3) were encoded as positive AKI. For ALI classification, we considered the presence of either NIV or CPAP during the admission, because PaO$_2$/FiO$_2$ measurements were not available. The ALI onset was the first occurrence of either NIV or CPAP (see Fig. 5).

**Prediction module.** The AI-EWS model is designed as a variation of a convolutional neural network (CNN) called a temporal convolutional network (TCN). CNNs have dominated computer vision tasks for the last century and are also highly capable of performing sequential tasks, such as text analysis and machine translation[41]. A TCN[23,24] models the joint probability distribution over sequences by decomposing the distribution over discrete time-steps $p_\theta(x) = \prod_{t=1}^{T} p_\theta(x_t|x_{1:t-1})$, where $x = \{x_1, x_2, \ldots, x_T\}$ is a sequence, and the joint distribution is parameterized by the TCN parameter $\theta$. Thus, a TCN operates under the autoregressive premise that only past values affect the current or future values, e.g., if a patient will develop acute critical illness. Moreover, TCNs differ from "ordinary" CNNs by at least one property: the convolutions in TCNs are causal in the sense that a convolution filter at time $t$ is only dependent on the inputs that are no later than $t$, wherein the input subsequence is $x_1, x_2, \ldots, x_t$. TCNs can take a sequence of any length as input and output a sequence of the same length, similar to RNNs[22,28]. The TCN achieves this by increasing the receptive field of the model with dilated convolutions instead of performing the traditional max pooling operation, as seen in most CNNs. Dilated convolutions achieve a larger receptive field with fewer parameters by having an exponential stride compared to the traditional linear stride. By increasing the receptive field, a temporal hierarchy comparable to multi-scale analysis from computer vision can be achieved[42]. Figure 6 schematizes the xAI-EWS model and the concept of dilated convolutions. At the time of prediction, the xAI-EWS model receives an input matrix of shape time-steps × features for each patient.

The data are processed by three temporal blocks, each including one-dimensional dilated causal convolutions (Conv1d) with 64 filters, ReLU activations[43], layer[44], and one-dimensional spatial dropout layers[45]. Dilation is increased between each temporal block, but keep it constant inside each temporal block (meaning that the second conv1d layer in each temporal block has a dilation = 1). The receptive field for this model can be calculated with $1 + \sum_{i=1}^{n}(k-1)*(2^{i-1} + 1)$, where $k$ is kernel size and $n$ is the number of temporal blocks. We used a kernel size = 4 yielding a maximum receptive field of 31. Outputs from the third temporal block are pooled together across time-steps by a global average pooling operation[46] to obtain a stabilizing effect for the final

output of the model. The pooled output from each kernel in the dilated causal convolutions is flattened to a single vector that is used as input to a final dense layer followed by a softmax activation function. The output from the softmax activation is the probability of future sepsis, AKI, or ALI during admission.

**Training and hyperparameters.** The model was trained to optimize the cross-entropy loss using the Adam optimizer[47] with mini-batches of the size of 200, a learning rate of 0.001, and a dropout rate of 10%. All weights were initialized with He Normal initialization[48]. The model was trained on a NVIDIA Tesla V100 GPU. Convergence was reached in ~30 min.

**Explanation module.** In simple models, such as linear regression models, the simple association between input and the prediction outcome is readily transparent and explainable. Consider the linear function $f_c$ that weights the input $\mathbf{x}$ by $w_c$ in order to assign a decision for class $c$:

$$f_c(\mathbf{x}) = \mathbf{w}_c^T x = \sum_i w_{ic} x_i. \tag{1}$$

Here, each input feature $x_i$ of $\mathbf{x}$ contributes together with the trainable weight $w_{ic}$ to the overall evaluation of $f_c$ through the quantity $w_{ic}x_i$. The importance-weighted input, therefore, offers a simple explanation for a decision made by the linear model. In contrast, the complexity associated with the multi-layer non-linear nature of deep learning models counteracts with such simplicity in explanations.

Layer-wise relevance propagation (LRP)[23–27] is an explanatory technique that applies to deep-learning models, including TCNs. Starting from the output $f_c(\mathbf{x})$, LRP decomposes an explanation into simpler local updates, each recursively defining the contribution to the explanation (called relevance) for all activating neurons in the previous layer. The initial relevance score $R_j = f_c(\mathbf{x})$ is hereby propagated backward through the network by local relevance updates $R_{i\leftarrow j}$ between connecting neurons $i$ and $j$, until the input layer is finally reached. In this process, all incoming relevance values to an intermediate node $i$ are pooled, $R_i = \sum_j R_{i\leftarrow j}$, before its relevance is propagated to the next layer. Figure 7 illustrates the relevance propagation, which is similar to standard backpropagation of errors except that relevance values are propagated backward in the network instead. The conservation property[23], one of the important defining properties in LRP, ensures that the total back-propagated relevance amounts to the extent to which the illness of interest is detected by the function $f_c(\mathbf{x})$, which in this paper equals the logits that feed into the final transformation layer.

In this process, all incoming relevance values to an intermediate node $i$ are pooled, $R_i = \sum_j R_{i\leftarrow j}$, before its relevance is propagated to the next layer. Figure 7 illustrates the relevance propagation, which is similar to standard backpropagation of errors except that relevance values are propagated backward in the network instead.

There are many variations of local backpropagation rules in the LRP framework. See, e.g., Montavon et al.[49] for a collection of commonly used LRP rules. We have used propagation rules that can be interpreted as DTD[23], which defines a sound theoretical framework behind most of the LRP variations. In DTD, a local backward propagation of relevance accounts for a non-linearity in the network model by a first-order Taylor approximation at some well-chosen root-point. Using the origin as root recovers the original LRP update rule from Bach et al.[25]. That is, the relevance $R_j$ at neuron $j$ propagates backward to neuron $i$ as

$$R_{i\leftarrow j} = \frac{w_{ij}a_i}{\sum_i w_{ij}a_i} R_j \tag{2}$$

where $a_i$ is the activation for neuron $i$. Notice that this local relevance-update rule is similar to the simple explanation for the linear model in Eq. (1), except that the normalization ensures that relevance is conserved across layers. It is the

**Table 3 Parameter range summary.**

| Parameter | Unit | Sepsis positive(5th/50th/95th percentile) | Sepsis negative(5th/50th/95th percentile) | AKI positive(5th/50th/95th percentile) | AKI negative(5th/50th/95th percentile) | ALI positive(5th/50th/95th percentile) | ALI negative(5th/50th/95th percentile) |
|---|---|---|---|---|---|---|---|
| Diastolic blood pressure | mmHg | 48/71/99 | 53/74/98 | 47/70/101 | 51/73/98 | 47.75/72/101 | 51/73/98 |
| Pulse rate | bpm | 63/90/127 | 56/77/107 | 59/82/120 | 56/79/111 | 61/94/131 | 56/78/111 |
| Respiration frequency | bpm | 12/19/32 | 12/16/22 | 12/18/28 | 12/16/24 | 14/20/38 | 12/16/24 |
| $SpO_2$ (Pulsoxymetry) | %$O_2$ | 88/95/99 | 92/97/100 | 90/96/100 | 91/97/100 | 85/94/99 | 91/97/100 |
| Systolic blood pressure | mmHg | 88/125/175 | 99/129/173 | 88/127/181 | 96/128/172 | 85/125/175 | 96/128/173 |
| Temperature | degrees (°) | 36.3/37.4/39.3 | 36.2/37/38.1 | 36.1/37.1/39 | 36.2/37.1/38.5 | 36.3/37.3/39.2 | 36.2/37.1/38.5 |
| P(aB)-Hydrogen carbonate | mmol/l | 17.4/24.2/31.595 | 18.9/25.2/33.7 | 15.2/22.9/30.94 | 17.5/24.2/31.8 | 17.3/24.1/31.8 | 17.3/24.9/33.9 |
| P(vB)-Hydrogen carbonate | mmol/l | 15.86/23.2/30.83 | 17/23.6/29.875 | 15.2/22.4/28.105 | 15.1/23/28.6 | 17.38/25.4/33.43 | 15.59/23/28.4 |
| P(aB)-$pO_2$ | kPa | 7.3/11/22.87 | 7/10.7/19.4 | 7.1/11.1/21.7 | 6.8/10.8/22.9 | 6.6/10.5/22.05 | 6.8/10.8/22.8 |
| P(vB)-$pCO_2$ | kPa | 4.315/6.3/8.17 | 4.1/5.8/7.9 | 3.2/5.3/9 | 3.5/5.3/9.1 | 5.1/6.8/10.11 | 3.7/5.5/7.4 |
| P(aB)-$pCO_2$ | kPa | 3.8/5.4/8.3 | 3.7/5.4/8.3 | 4.23/6.3/7.9 | 3.6/5.5/7.7 | 3.8/5.5/9.4 | 3.5/5.3/9.2 |
| P(aB)-pH | – | 7.23/7.4/7.5 | 7.29/7.42/7.51 | 7.2/7.37/7.49 | 7.25/7.4/7.5 | 7.22/7.4/7.5 | 7.24/7.4/7.5 |
| P(vB)-pH | – | 7.21/7.34/7.44 | 7.21/7.36/7.46 | 7.19/7.33/7.45 | 7.2/7.37/7.49 | 7.19/7.37/7.45 | 7.21/7.36/7.45 |
| P(aB)-Lactate | mmol/l | 0.6/1.4/5.68 | 0.6/1.3/3.6 | 0.6/1.4/5.29 | 0.6/1.4/5 | 0.6/1.4/5 | 0.6/1.3/4.3 |
| P(aB)-Sodium | mmol/l | 130/138/148 | 130/138/146 | 129/137/144 | 128/137/144 | 130/138/147 | 128/137/144 |
| P(aB)-Chloride | mmol/l | 96/105/113 | 95/105/113 | 96/107/115 | 95/105/113 | 94/104/115 | 95/105/113 |
| P(aB)-Potassium | mmol/l | 3.2/4/5.2 | 3.3/4/4.9 | 3.3/4.2/5.6 | 3.2/4/5 | 3.2/4/5.3 | 3.2/4/5.1 |
| B-Hemoglobin | mmol/l | 4.9/6.9/9.5 | 5.3/7.7/9.7 | 5.3/7.7/9.7 | 5.3/7.7/9.7 | 4.9/7/9.5 | 5.3/7.7/9.7 |
| B-Neutrophils | $\times 10^9$/l | 0.15/3.7/15.51 | 0.37/5.18/14.75 | 0.14/5.09/27.5 | 0.25/4.825/14.6 | 5.162/6.05/10.19 | 0.26/4.76/15.26 |
| B-Erythrocyte sedimentation rate | mm | 5.35/50/117 | 2/19/98.09 | 5/36/108.9 | 3/19/99 | 3/28.5/98.3 | 3/19/98.70 |
| B-Platelets | $\times 10^9$/l | 26/184/429 | 94/229/416 | 76.45/218/426 | 88/229/431 | 59/203/475.55 | 88/228/428 |
| B-Leukocytes | $\times 10^9$/l | 2.13/11.1/24.96 | 4.15/8.53/16.8 | 4.53/9.82/20.4 | 4.1/8.97/18.8 | 4.65/11.4/26.52 | 4.14/8.87/18.6 |
| P-Sodium | mmol/l | 128/138/146 | 132/139/144 | 129/138/145 | 131/139/144 | 130/139/147 | 131/139/144 |
| P-Potassium | mmol/l | 3/4/5.1 | 3.1/3.9/4.7 | 3.1/4.1/5.4 | 3.1/3.9/4.7 | 3/4/5.1 | 3.1/3.9/4.8 |
| P-Lactate dehydrogenase (LDH) | U/l | 128/195/491 | 150/262/1060.4 | 140/222/552.95 | 128/197/505 | 139.4/262/1023 | 128/197/511 |
| P-Albumin | µmol/l | 25/37/46 | 25/37/46 | 23/35/44 | 26/37/46 | 19/30/41 | 26/37/46 |
| P-Creatinine | µmol/l | 40/95/359 | 44/75/188 | 55/124/662 | 44/76/182 | 36/81/307.9 | 44/77/210 |
| P-Bilirubin | µmol/l | 3/8/35 | 4/12/98.25 | 3/8/40 | 3/8/40 | 3.55/10/78 | 3/8/40 |
| P-Prolactin | $\times 10^3$/l | 786/786/786 | 81.2/345/1976.4 | – | 3/8/40 | – | 84/347/1790 |
| P-Glucose | mmol/l | 4.44/8.9/21.13 | 4.5/8.5/18.5 | 3/8/40 | 3/8/40 | 4.3/9.4/20.3 | 4.5/8.5/18.8 |
| P-C-reactive protein (CRP) | mg/l | 3.85/79.3/328.18 | 1.4/26.6/199.4 | 1.5/29.2/226.9 | 1.5/29.2/226.9 | 4.3/77.5/352.9 | 1.5/28.4/225 |
| Hb(B)-Hemoglobin A1c | mmol/mol | 30.65/40.5/68.1 | 31/39/71 | 31/43/77 | 31/39/73 | 31.3/39.5/70.95 | 31/39/73 |
| Glomerular filtration rate (eGFR) | ml/min | 10/45/85 | 18.9/25.2/33.7 | 6/40/82 | 22/66/88 | 18/64/88 | 13/52/88 |

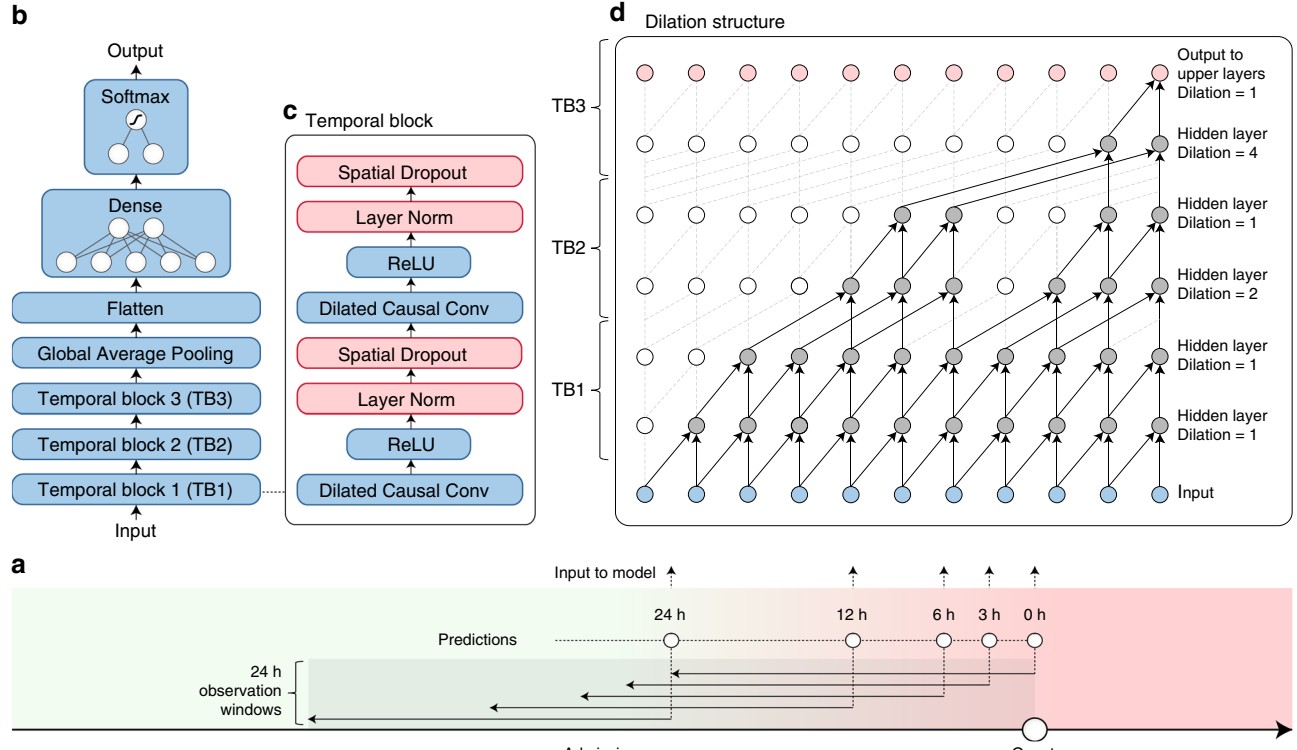

**Fig. 5 Gold standards for sepsis, AKI, and ALI.** Sepsis (**a**), AKI (**b**), ALI (**c**), suspected infection (SI).

**Fig. 6 The xAI-EWS model architecture.** The models in this study are trained and evaluated at 0, 3, 6, 12, and 24 h before the onset of critical illness. Each model has a 24-h retrospective observation window. The color gradient from green to red illustrates continuous deterioration towards acute critical illness (**a**). The overall model architecture of the AI-EWS model is shown in **b**. The xAI-EWS uses three temporal blocks (**c**), each comprising one-dimensional dilated causal convolution layers, ReLU activations, one-dimensional dropout layers, and normalization layers. Red layers are only used during training and are omitted when the model is used for predictions and explanations. The overall dilation structure of the model is shown in **d**. The one-dimensional dilated causal convolution layers allow the model to skip some points during convolution and, thereby, increase the receptive field of the model. The dilation structure is illustrated for kernel size = 2.

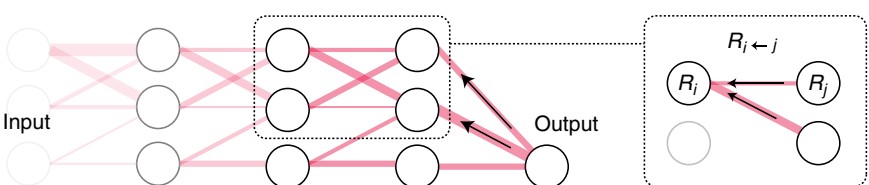

**Fig. 7 Layer-wise relevance propagation.** LRP decomposes the problem of explaining a complex multilayer neural network model into simpler sub-functions that are easier to analyze and explain. The relevance score at input neuron $R_i$ is obtained by pooling all incoming relevance values $R_j$ from the output neurons in the next layer.

simplest local relevance propagation in the LRP framework, known as simply LRP, LRP-0, or the z-rule in DTD.

In general, the Taylor expansion that defines the local relevance propagation rule depends on the type of non-linearity and can, in addition, be engineered to enforce desirable properties based on root-point restrictions[23]. The network for the model considered in this work is composed of only ReLU activations and linear projections without a bias term. In this case, we can use a particularly engineered

rule that will only distribute relevance along positive contributions through the layers in the network and therefore produces sparser (i.e., simpler) explanations. This rule is in the literature known as the z+-rule for DTD or LRP-$\alpha_1\beta_0$, which again is a special case of LRP-$\gamma$[49]. It is defined as

$$R_{i \leftarrow j} = \frac{w_{ij}^+ a_i}{\sum_i w_{ij}^+ a_i} R_j \qquad (3)$$

where $w_{ij}^+ = w_{ij}$ for positive weights and otherwise equals zero. Finally, at the input layer, we used the so-called DTD $w^2$-rule

$$R_{i \leftarrow j} = \frac{w_{ij}^2}{\sum_i w_{ij}^2} R_j \qquad (4)$$

as it is recommended[49] for real valued input.

The DTD (and LRP) framework leaves flexibility to mix layer-specific rules in the network. As mentioned above, we have used the $z^+$-rule in Eq. (3) for all intermediate layers, and in that way, we favor simpler explanations, with features that are explained as either relevant (positive) or irrelevant (zero) for a given prediction.

The AI-EWS explanation module allows two perspectives on model explanations: an individual and a population-based perspective. For the individual perspective, DTD can be used for all patients with a high probability of developing acute critical illness. The module will simply pinpoint which clinical parameters at a given point in time were relevant for the given prediction (Fig. 3). For the population-based perspective, relevance is back-propagated from the output neuron representing the positive classes (sepsis, AKI, and ALI) and is only considered for the patients with a positive ground truth label (sepsis, AKI, and ALI). The individual data points and back-propagated relevance scores for these patients were aggregated in two ways to enable global parameter importance estimation and local explanation summary[34] (Fig. 4). For estimating global parameter importance, the mean relevance scores were computed for each clinical parameter. This computation enabled parameter-importance estimation comparable to standardized regression coefficients in multiple linear regression or feature importance measures in random forest[50]. The local explanation summary (Fig. 4b) presents all individual data points, colored by parameter value and displaced by the relevance. In the local explanation summary, the height of the data points shown for each parameter correlates with the number of data points at their associated level of relevance. The population-based perspective is simplified by ignoring potential temporal relevance variations and treating all data points at different times equally. The visual concepts of global parameter importance estimation and local explanation summary used in this paper are adopted from the shapley additive explanations (SHAP)[34] library by Lundberg et al. The SHAP toolbox was not used to provide explanations. In this paper, DTD was implemented using the iNNvestigate[51] library developed by Alber et al. iNNvestigate is a high-level library with an easy-to-use interface for many of the most-used explanation methods for neural networks.

**Explaining predictions in other ways**. Over the last decade, many other methods[52–63] have been proposed to address the problem of attributing a value to each feature in order to explain the prediction from a complex model. Recent work[62,63] have brought some theoretical understanding into fundamental properties that relates many of the methods. In general, the relevance attribution methods can be categorized into two basic categories[62]: (1) backpropagation-based methods that propagate the attribution of relevance backward through the network form the relevance of output and back to the input features, and (2) perturbation-based methods that rely on running multiple perturbations of the input forward through the network and measuring the consequence that a perturbation may have on the output.

With reference in the Gradient × Input method[58], Ancona et al.[62] defines a unification of many of the backpropagation-based attribution methods. In particular, they show that LRP (DTD with the z-rule), DeepLIFT (Rescale)[59], and Integrated Gradients[60] are all strongly related by a reformulation that expresses the attribution as the elementwise multiplication of a modified gradient with either the input (Gradient × Input, LRP) or with the difference between the input and a baseline (Integrated Gradients, DeepLIFT). The modification to the gradient is achieved by letting backpropagation implement a modified chain-rule

$$\frac{d^* a_j}{da_i} = w_{ij} g(z_j), \qquad (5)$$

where $w_{ij}$ is the standard gradient of the linear transformation $z_j = \sum_i w_{ij} a_i$, and $g(z_j)$ represents some unification of the gradient for the non-linear activation. For Gradient × Input, the unifying gradient function $g(z_j)$ is the usual instant gradient, whereas it is some form of average gradient for the remaining three methods. We refer to Ancona et al. for the actual expressions of $g(z_j)$. Interestingly, it turns out that all four methods would be equivalent in this paper's setup, with only ReLU non-linearities, no additive bias in the linearities, and $\mathbf{x} = \mathbf{0}$ as baseline.

Now, the DTD $z^+$-rule, as we have used in all the intermediate layers, does not fit directly into the unified gradient framework, but it is a trivial exercise to show that the backpropagation rule in Eq. (5) can be modified as

$$\frac{d^* a_j}{da_i} = w_{ij}^+ g^+(z_j), \qquad (6)$$

where $g^+(z_j) = a_j / z_j^+$, $z_j^+ = \sum_i w_{ij}^+ a_i$ accounts for the fact that the relevance method only distributes relevance along positive contributions.

Backpropagation-based methods are in general fast compared to perturbation-based methods, as the number of features grow[62]. The backpropagation-based methods require a fixed number of forward-prediction and backward-gradient passes to compute the attribution, whereas perturbation-based methods demand a non-linear number of forward passes in the number of features to properly account for the complex nature of a deep network.

On the other hand, perturbation-based methods may implement other desirable properties, such as the fairness constraints from cooperative game theory, when attributing an outcome of a prediction (the game) to the individual features (the players)[53,61,63,64]. In particular, Lundberg and Lee[63] recently demonstrated that Shapley values[65] uniquely defines the solution to these constraints within a large class of additive feature attribution methods, which includes LIME[57], DeepLIFT, and LRP. Unfortunately, computing exact Shapley values is, in general, NP-hard[66] and sampling approximations are therefore considered. By defining a specific kernel in the LIME setup, Lundberg and Lee introduced KernelSHAP that reduces the number of necessary samples by combining sampling and penalized linear regression, as it is done in LIME. The same paper further proposed DeepSHAP as a variant of DeepLIFT that computes a layer-wise composition of approximate Shapley values. Unfortunately, the chain rule does not hold in general for Shapley values[64], and to that end, Ancona et al. presents a method based on uncertainty propagation that allows approximate Shapley value to be computed in polynomial time.

**Baseline models**

*MEWS*. The MEWS baseline model interprets raw MEWS scores as a prediction model for acute critical illness. MEWS was implemented as the Danish variant called "Early detection of critical illness" [TOKS: Tidlig opsporing af kritisk sygdom]. MEWS scores were calculated each time one of the model components was updated with a new measurement. Missing values were imputed with a standard carry-forward interpolation.

*SOFA*. This model interprets raw SOFA[30,39,40] scores as a prediction model for acute critical illness. SOFA scores were calculated each time one of the model components was updated with a new measurement. Missing values were imputed with a standard carry-forward interpolation.

*GB-Vital*. This model is a replication of a well-known sepsis detection model[3,6,9] from the literature, which has shown great results in a randomized study[7]. The complete description of the model can be found in the study from Mao et al.[6]. The model parameters are constructed by considering six vital-signs from the EHR: systolic blood pressure, diastolic blood pressure, heart rate, respiratory rate, peripheral capillary oxygen saturation, and temperature. For each of the six vital signs, five parameters are constructed to represent the average value for the current hour, the prior hour, and the hour prior to that hour, together with the trend value between two succeeding hours. Based on these 30 parameters (five parameters from each of the six vital-sign events), the GB-Vital model is constructed as a gradient-boosted classifier of decision trees[67].

**Evaluation**. The xAI-EWS model was validated using five-fold cross-validation. Data were randomly divided into five portions of 20% each. For each fold four portions (80%) was used to fit the xAI-EWS model parameters during training. The remaining 20% was split into two portions of 10% each for validation and test. The validation data were used to perform an unbiased evaluation of a model fit during training, and the test data were used to provide an unbiased evaluation of the final model. All data for a single patient was assigned to either train, validation or test data. Figure 2 report performance from the test data. For each fold data were shifted such that a new portion was used for testing. The cross-validation scheme is illustrated in Supplementary Fig. 1. As comparative measures for the predictive performance, we used the AUROC and AUPRC. Regarding the explanations, the quality was assessed by manual inspection of trained specialists (medical doctors) in emergency medicine.

**Ethics and information governance**. The study was approved by The Danish Data Protection Agency [case number 1-16-02-541-15]. Additionally, the data used in this work were collected with the approval of the steering committee for CROSS-TRACKS. Only retrospective data were used for this research without the active involvement of patients or potential influence on their treatment. Therefore, under the current national legislature, no formal ethical approval was necessary.

**Reporting summary**. Further information on research design is available in the Nature Research Reporting Summary linked to this article.

## Data availability

The authors have accessed the data referred to herein by applying the CROSS-TRACKS cohort, which is a newer Danish cohort that combines primary and secondary sector data[68]. Due to the EU regulations, GDPR, these data are not readily available to the wider research community per se. However, all researchers can apply for access to the data by following the instructions on this page: http://www.tvaerspor.dk/.

## Code availability

We made use of several open-source libraries to conduct our experiments: The models used the machine learning framework TensorFlow library with custom extensions (https://www.tensorflow.org) and Keras (https://keras.io). Explanations were calculated with the high-level library for explaining neural networks iNNvestigate (https://github.com/albermax/innvestigate). SHAP (https://github.com/slundberg/shap) with custom extensions was used to visualize explanations. The analysis was performed with custom code written in Python 3.5. Our experimental framework makes use of proprietary libraries that belong to Enversion A/S, and we are unable to publicly release this code. We have described the experiments and implementation details in the "Methods" section to allow for independent replication. Further inquiry regarding the specific nature of the AI model can be made by relevant parties to the corresponding author.

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

## Acknowledgements

We acknowledge the steering committee for CROSS-TRACKS for access to the data. For data acquisition, modeling, and validation, we thank the following: Julian Guldborg Birkemose, Christian Bang, Per Dahl Rasmussen, Anne Olsvig Boilesen, Lars Mellergaard, and Jacob Høy Berthelsen. For help with the data extraction pipelines, we thank Mike Pedersen. We also thank the rest of the Enversion team for their support. This work was also supported by the Innovation Fund Denmark [case number 8053-00076B].

## Author contributions

S.M.L., J.L., M.J.J., and B.T. initiated the project. B.T., S.M.L., K.M.L., J.L., and M.J.J. contributed to the overall experimental design. S.M.L. and M.K. created the dataset. S.M.L., M.K., M.V.O., and M.S.L. contributed to the software engineering and K.M.L., B.T., and S.M.L. analyzed the results. S.M.L. made the first article draft. All authors contributed significantly to revision of the first article draft, and approval of the final version of the manuscript.

## Competing interests

S.M.L, M.K., and B.T. are employed at Enversion. The authors have no other competing interests to disclose.
