## [Peer Review File · Nature Communications]

Reviewers' Comments:

Reviewer #1:

Remarks to the Author:

This work presents an interpretable deep learning framework for the continuous prediction of sepsis, acute kidney injury, and acute lung injury based on hourly electronic health records data and temporal convolutional networks. As stated by the authors, the primary aim was to demonstrate explainable AI in healthcare using deep Taylor decomposition – as opposed to superlative prediction performance – although their methods also quantitatively outperform all three included baseline models. Overall, the manuscript is very well-written, understandable, visually appealing, concise to the point of the authors' primary objectives, and contains mostly satisfactory technical explanation in the extended material.

While I enjoyed the manuscript and feel it eventually deserves to be published, below are several comments that I feel – once addressed – will lead to a stronger publication. They are primarily clarification questions that I believe the authors should address in further detail.

- I believe there should be more discussion on alternative input-based saliency techniques (e.g. DeepLIFT, SHAP, integrated gradients, etc.) and why deep Taylor decomposition was chosen as the technique presented in this manuscript. Further, there should be a more detailed description of related work in this area, as applying these techniques to healthcare data is not exactly novel (the authors cite many studies, but I think a few of the most related could be explicitly and briefly mentioned)

- On page 2, the authors state that explanation "quality was assessed by manual inspection of trained specialists (medical doctors) in emergency medicine", but this is not mentioned again or expounded upon for the remainder of the manuscript. Did clinicians agree or disagree with, did they find them helpful, etc.? These are very important questions for a study like this.

- Since the authors themselves list the ability to handle variable length sequences as one of the two defining characteristics of a TCN, it seems odd to instead use fixed length sequences of 24 hours in the experiments. The authors should briefly mention why full time series (of variable length) were not used in favor of (essentially) their sliding window approach.

- For global feature importance, the authors mention "only considering for the patients with a positive label". It should be clarified whether they are referring to the ground truth or predicted label.

- Also regarding global importance, it appears that per-feature means are used to summarize overall importance. Since objective relevance values are based on the total model output, could this potentially miss important features for the most difficult cases to predict (where the total output to be propagated is low)?

- Minor point of contention: I would argue that while interpretability is a good thing, the following statements about the necessity of interpretability in medicine are somewhat debatable and dependent on context (for example, one might prefer to exchange transparency for predictive power in either low stakes settings or if the benefit is large enough): "the importance of explainable and transparent DL algorithms is without question", "transparency and explainability are an absolute necessity"

- The authors briefly mention receptive field, but it should be explicitly stated using given hyperparameters, whether the receptive field at the final TCN layer extends to cover the entire input sequence (as with RNNs)

- A variable summary table would be nice to see (value ranges, measurement frequency, etc. between positive and negatively labelled patient groups).

- The outcome prevalence rates are very low (2.44%, 0.75%, 1.68%). The authors should explain whether any techniques were used to account for this (loss weighting, resampling, etc.)

- In Figure 2, how was "mean" AUROC/AUPRC calculated? I would guess once per hour over 24 hours, but this should be briefly clarified.

- Given the authors' discussions of sensitivity and specificity tradeoffs in baseline models, it would be nice to see sensitivity and specificity results for their framework as well. I would argue that these statistics could be equally important as feature attribution assignment for clinician trust.

- Given (1) the authors' focus on critical illness, (2) the hospital mortality statistic presented in Extended Table 2, and (3) the basis target for the SOFA score being in-hospital mortality, it is a little perplexing why in-hospital mortality was not included as an outcome of interest. I think the rationale behind choosing outcomes should be expanded upon.

- How can raw relevance scores be interpreted in isolation, if at all? For example, what does a relevance score of 0.4 mean, and how does it compare to one of 0.3? Do relevance scores across input features sum to the overall prediction probability? A brief description would be beneficial, as currently the assessment of important features appears to be in relation to other features from the same input.

- It is nice to see the authors list hyperparameters, however I could not find a crucial one: TCN kernel size/width. This should be listed as well.

- I take slight issue with the authors definition of a TCN, with one of the two hallmarks being that "TCNs can take a sequence of any length as input and output a sequence of the same length". I would argue that while this is indeed how TCNs can work, it is not a defining characteristic (since one may also desire this property in "standard" CNNs, without requiring causality) and is more related to the causal aspect of definition #1.

Reviewer #2:

Remarks to the Author:

Title of the Paper: Explainable artificial intelligence model to predict acute critical illness from electronic health records

This paper reports on the application of an explainable ai method (namely Layer-wise relevance propagation (LRP)) on a neural network, which was conceived to perform a prediction from temporal data. The data, preprocessing steps, neural network architecture, as well as the results of the relevance of each feature, are presented.

1) Originality:

The novel aspect is the domain that is used; because the medical domain is a very important domain. Other than that, there are a lot of aspects that the paper should have reasoned about; but the application domain justifies originality

2) Related Work:

Very important is that the authors mention (maybe in a sentence after line 42, page 1, or at the end) that explainability as it is described here is a technical precondition, but for a complete "explanatory loop" useful for the practical (!) medical domain expert the aspect of measuring how good an explanation fits to a certain problem and a certain domain expert (to whom it may be explained); consequently the authors should mention the concept of causability [xx] and how to measure causability [yy]:

[xx] Holzinger, A., Langs, G., Denk, H., Zatloukal, K. & Mueller, H. 2019. Causability and Explainability of Artificial Intelligence in Medicine. Wiley Interdisciplinary Reviews: Data Mining and Knowledge Discovery, 9, (4), (2019) doi:10.1002/widm.1312.

[yy] Holzinger, A., Carrington, A. & Müller, H. 2020. Measuring the Quality of Explanations: The System Causability Scale (SCS). Comparing Human and Machine Explanations. Springer/Nature KI - Künstliche Intelligenz (German Journal of Artificial intelligence), Special Issue on Interactive Machine Learning, Edited by Kristian Kersting, TU Darmstadt, 34, (2), doi:10.1007/s13218-020-00636-z.

3) Methodology:

The LRP method is briefly covered and in general the research work as it is described is correct. However, there are some important aspects missing, like e.g., perturbation analysis (masking tests). This method is contained in the investigate tool for the images, but should also be applied in this case, especially when the researchers state in page 2 that they wish to show robustness or in page 4 to show which parameters were the causes. This would give them much more insights than the (awaited) interpretation: The nearer to the disease time, the more relevant a feature is: "more weight to recent values" p.4. LRP papers go beyond that and show how the class prediction is affected by the gradual removal of the input features (pixels, words – word ablation) that are relevant and furthermore compare that to other eXAI methods. One important constraint that needs to be ensured for the application of this method is that the bias parameters are negative – this also can produce a negative effect in the performance of the NN. This is not mentioned and cannot be verified from the github repository as it is right now (this reviewer carefully checked this).

4) Results:

The results are clearly presented, but there is one exception: In p. 15 the baseline models are referred, but how are they related to the NN? LRP papers explain the dependency of a good explanation from the performance of the model.

5) Qualitative Evaluation:

The paper is well written, easy to read and the figures are of good quality. Particularly p.6 is well written. Some problems occur in p.14 where the SAHPs are explained. It is not clear to the reviewer i.e. what the connection between the population-based LRP case has to do with Shapely values? A minor correction could be the use of the term "parameter" instead of "input feature" – "parameter" is usually used instead of neural network parameter.

6) Summary and Recommendation:

The paper is interesting, relevant and well written. This reviewer would recommend that the authors do some minor revisions as outlined above. The authors can be trusted to do these revisions on their own; btw. in p.10 "known correlations" are referred. Is it possible that the authors measure those correlations (non-linear)? Can there be overlaps between the input features?

Accept with minor revisions

Dear Reviewers

We thank you for your comments and feedback.

We have taken the liberty of dividing the comments into separate parts. Each part is seen in the table below with our comments and changes in the revised manuscript.

Best Regards

The authors

Number	Reviewer comments	Author comments	Change in revised manuscript
Reviewer 1			
1.1	This work presents an interpretable deep learning framework for the continuous prediction of sepsis, acute kidney injury, and acute lung injury based on hourly electronic health records data and temporal convolutional networks. As stated by the authors, the primary aim was to demonstrate explainable AI in healthcare using deep Taylor decomposition – as opposed to superlative prediction performance – although their methods also quantitatively outperform all three included baseline models. Overall, the manuscript is very well-written, understandable, visually appealing, concise to the point of the authors' primary objectives, and contains mostly satisfactory technical explanation in the extended material.	We thank the reviewer for the appreciation of our paper.	No changes made in the document.
1.2	While I enjoyed the manuscript and feel it eventually deserves to be published, below are several comments that I feel – once addressed – will lead to a stronger publication. They are primarily clarification questions that I believe the authors should address in further detail.	We are glad that the reviewer finds our work suitable for publication in Nature Communications and hope that our corrections have led to a better and clearer work.	No changes made in the document.
1.3	- I believe there should be more discussion on alternative input-based saliency techniques (e.g. DeepLIFT, SHAP, integrated gradients, etc.) and why deep Taylor decomposition was chosen as the technique presented in this manuscript.	We fully acknowledge the point of view made by the reviewer with respect to alternative methods. We have therefore added a new section "Explaining predictions in other ways" to discuss alternative	Added new section "Explaining predictions in other ways". Changed Section "Explanation module" by: 1) adding "vanilla" LPR description (the DTD z-rule) – Eq. (2a) and surrounding text.

	Further, there should be a more detailed description of related work in this area, as applying these techniques to healthcare data is not exactly novel (the authors cite many studies, but I think a few of the most related could be explicitly and briefly mentioned)	saliency attribution methods (Gradient x Input, Integrated Gradients, DeepLIFT, “vanilla” LRP, LIME, and SHAP + variations) and how they relate. In our particular setup, it turns out that many of these methods are, in fact, equivalent for all the intermediate layers of our network - as we describe in that section as well. (Our model only has ReLU non-linearities, no additive bias in the linearities, and we use $x=0$ as data baseline) We have also tightened up the original description of LRP (and the DTD interpretation) to better appreciate how the specific propagation rule that is used in the paper differs from “vanilla” LRP (the so-called LRP-0). The difference lies in the fact that relevance is only distributed along positive contributions through the layers in the network – the effect being that we prefer sparser (i.e. simpler) explanations. Finally, we have also added the rule used at the final relevance propagation step to the input layer to ensure complete reproducibility of our approach. As pointed out by the reviewer there should be a more detailed description of related work in this area, as applying these techniques to healthcare data. We have added such a section to the discussion.	2) better explaining the LRP approach (DTD z^+-rule) that we use for all intermediate layers and its property of producing sparser explanations – text surrounding Eq. (2b). 3) Added the DTD-w^2 propagation rule for the input layer – Eq. (2c) and surrounding text. Added section about related work: “Previous work has employed different strategies to develop explainable prediction models^{15,34–38}. RNN variations with attention have been suggested for illness severity assessment¹⁵, risk of hospitalization prediction³⁴, sepsis prediction, and myocardial infarction prediction³⁷. Shickel et al. developed an interpretable deep learning framework called DeepSOFA that leveraged temporal measurements to assess illness severity at any point during an ICU stay. An RNN with gated recurrent units (GRU) and self-attention was proposed to highlight particular time steps of the input time series that the model believed to be most important in formulating its mortality prediction. Kaji et al. demonstrated how attention can be applied at the level of input variables themselves when predicting outcomes for ICU patients³⁷. Choi et al. used a factorized approach to compute attention over both variables and time using embedded features rather than the immediate input features themselves³⁸. Zhang et al. compressed the entire patient EHR into a complete vector representation and used GRU and self-attention to predict the future risk of hospitalization in an interpretable framework called Patient2Vec³⁴. Our work differentiates from above studies by utilizing TCN and LRP instead of RNNs with attention.”
--	---	---	--

1.4	- On page 2, the authors state that explanation “quality was assessed by manual inspection of trained specialists (medical doctors) in emergency medicine”, but this is not mentioned again or expounded upon for the remainder of the manuscript. Did clinicians agree or disagree with, did they find them helpful, etc.? These are very important questions for a study like this.	The development of the models in this study was done in an iterative way where results from technical development were continuously presented to, and discussed with, clinicians from an emergency department. The primary purpose of this iteration process was to ensure that the models learned at least some correlations that are already considered established knowledge in the clinical field. It would be obvious to try to use this technology hypothesis-generating, whereby output from LRP analysis is used as inspiration to discover new and unknown correlations. However, this has not been the purpose of the present work. In relation to inter- and intrarater variability (agree/disagree) in the next fase of the clinical translation of the algorithm this will be sufficiently evaluated for real-life performance of the algorithm.	No changes made in the document.
1.5	- Since the authors themselves list the ability to handle variable length sequences as one of the two defining characteristics of a TCN, it seems odd to instead use fixed length sequences of 24 hours in the experiments. The authors should briefly mention why full time series (of variable length) were not used in favor of (essentially) their sliding window approach.	We limited the length of our observation window to 24 hours to ensure that the model was based on clinical, and time-relevant, features. A variable length window greater than 24 hours should be explored in an upcoming study. In such a future study one should also investigate how explanations, as well as the consistency of the explanations, are influenced by how observation window size.	The following sentence was added in the end of the discussion: “We limited the length of the observation window to 24 hours to ensure that the model was based on clinical, and time-relevant, features. A variable length window greater than 24 hours should be explored in an upcoming study.”
1.6	- For global feature importance, the authors mention “only considering for the patients with a positive label”. It should be clarified whether they are referring to the ground truth or predicted label.	This refers to ground truth label, and has now been updated in the manuscript.	The statement now reads: “For the population-based perspective, relevance is back-propagated from the output neuron representing the positive classes (sepsis, AKI, and ALI) and is only considered for the patients with a positive ground truth label (sepsis, AKI, and ALI).”
1.7	- Also regarding global importance, it appears that per-feature means are	As pointed out by the reviewer it is true that relevance values are	No changes made in the document.

	used to summarize overall importance. Since objective relevance values are based on the total model output, could this potentially miss important features for the most difficult cases to predict (where the total output to be propagated is low)?	based on total model output, and this, along with the selected aggregation techniques, will have both advantages and disadvantages. When we use “per-feature means” as the aggregation method, we divide by the number of times a feature occurs. In this way, important features that are only rarely registered will still have the opportunity to be included in the top list. If we instead simply summarized the relevance, the top list would largely reflect how often the individual features are measured, rather than the importance. It is true that both of the above aggregation techniques do not potentially capture the important features of cases with low model output, thereby low back propagated output.	
1.8	- Minor point of contention: I would argue that while interpretability is a good thing, the following statements about the necessity of interpretability in medicine are somewhat debatable and dependent on context (for example, one might prefer to exchange transparency for predictive power in either low stakes settings or if the benefit is large enough): “the importance of explainable and transparent DL algorithms is without question”, “transparency and explainability are an absolute necessity”	The reviewer has a valid point. We agree and have tempered the statement.	The statement now reads: “A tradeoff must, therefore, be made between transparency and predictive power, which for high-stake applications most often falls out in favor of the simpler more transparent systems, where a clinician can easily trace back a prediction to the cause. To benefit from the higher predictive power, the importance of explainable and transparent DL algorithms in clinical medicine is therefore without question, as was also recently highlighted in the Nature Medicine review by Topol, E. J.^{27,28.}”
1.9	- The authors briefly mention receptive field, but it should be explicitly stated using given hyperparameters, whether the receptive field at the final TCN layer extends to cover the entire input sequence (as with RNNs)	We agree with the reviewer. According to “Extended Data Figure 2”, now called “Figure 6” we have two Conv1d layers in each temporal block. We increase the dilation rate between each temporal block, but keep it constant inside each temporal block (meaning that the second conv1d layer in each temporal block has a dilation=1). The	We updated the manuscript we these detail, and the sentence now reads: “The data is processed by three temporal blocks, each including one-dimensional dilated causal convolutions (Conv1d) with 64 filters, ReLU activations⁴³, layer⁴⁴, and one-dimensional spatial dropout layers⁴⁵. Dilation is increased

		receptive field can thus be calculated with the given formula: $1 + \sum_{i=1}^n (k - 1) * (2^{i-1} + 1)$ Where k is kernel size and n is the number of temporal blocks. We experimented with kernel sizes from 3-8. Kernel size did not impact performance greatly, and we ended up using a kernel size of 4 in the final experiments, yielding a receptive field of 31. $1 + \sum_{i=1}^n [(k-1)*(2^{i-1} + 1)] = 31$	between each temporal block, but keep it constant inside each temporal block (meaning that the second conv1d layer in each temporal block has a dilation=1). Receptive field for this model can be calculated with $1 + \sum_{i=1}^n (k - 1) * (2^{i-1} + 1)$, where k is kernel size and n is the number of temporal blocks. We used a kernel size=4 yielding a maximum receptive field of 31. Outputs from the third temporal block are pooled together across time-steps by a global average pooling operation⁴⁶ to obtain a stabilizing effect for the final output of the model.”
1.10	- A variable summary table would be nice to see (value ranges, measurement frequency, etc. between positive and negatively labelled patient groups).	We fully acknowledge the need for a summary table. We have added a new table (table 3) showing 5 th , 50 th , and 95 th percentiles for all variables.	We have added a variable summary table as table 3.
1.11	- The outcome prevalence rates are very low (2.44%, 0.75%, 1.68%). The authors should explain whether any techniques were used to account for this (loss weighting, resampling, etc.)	We experimented with resampling methods, specifically we tried oversampling for the positive class. The oversampling did not affect model performance, but it did affect the range of output probabilities. When we did not use oversampling the output probabilities were compressed to a smaller region of the probability space, such as 0 to 0.1. Oversampling affected this by increasing the upper limit from 0.1 to a point nearer 1.	We added the following section at the end of the discussion: “The low prevalence of the sepsis, AKI and ALI (2.44%, 0.75%, 1.68%) resulted in a very unbalanced classification problem. To combat this imbalance, we tried to oversample the positive class with replacements. The oversampling did not affect model performance significantly, but affected the stretch of output probabilities. When we did not use oversampling the output probabilities were compressed to a smaller region of the probability space, such as 0 to 0.1. Oversampling affected this by increasing the upper limit from 0.1 to a point nearer 1. The results reported were computed without resampling of any kind.”
1.12	- In Figure 2, how was “mean” AUROC/AUPRC calculated? I would guess once per hour over 24 hours, but this should be briefly clarified.	Mean values and 95% Confidence Intervals are calculated over the over the five cross-validations folds. We have updated the text to make this clear.	The statement now reads “AUROC with mean values and 95% Confidence Intervals (CIs) over the five cross-validations folds were 0.92(0.9-0.95)–0.8(0.78-83),...”

1.13	- Given the authors' discussions of sensitivity and specificity tradeoffs in baseline models, it would be nice to see sensitivity and specificity results for their framework as well. I would argue that these statistics could be equally important as feature attribution assignment for clinician trust.	We thank the reviewer for this comment, and agree in the point about the importance of these statistics. However, due to the following two reasons, we would rather not add sensitivity and specificity results to the current results: 1) In order to be able to report sensitivity and specificity, we are forced to choose one specific operating point, for the different models. The operating point, and thus also the tradeoff between sensitivity and specificity, is very subjective and depends to a large extent on how the model is to be used and implemented. For example, if the model is to be decision-supportive, one will typically choose a different operating point than if the model is to be used in a more automated manner. 2) This article, in its current form, presents a lot of results because we report on three different outcomes. As it can be seen from the AUROC/AUPRC results in figure 2 our model outperformed the clinical baseline methods. And, if a model dominates the entire the ROC-or PR-space, it will also offer a better tradeoff between sensitivity and specificity for all possible operating points. We will without a doubt pay much attention to the clinician trust to ensure optimal clinical translation in the next phases of implementation. Which include the specified statistics.	No changes made in the document.
1.14	- Given (1) the authors' focus on critical illness, (2) the hospital mortality statistic presented in Extended Table 2, and (3) the basis target for the SOFA score being in-hospital mortality, it is a little perplexing why in-hospital mortality was not included as an outcome of interest. I think the rationale behind	We absolutely agree that in-hospital mortality is an interesting endpoint to look at. In this study, we have chosen to look at the exacerbation of disease (acute critical illness) within some of the most frequent hospitalization causes, but it is obvious that further work focuses on mortality,	No changes made in the document.

	choosing outcomes should be expanded upon.	both as an independent endpoint, but also as a sub-label among the other diseases.	
1.15	- How can raw relevance scores be interpreted in isolation, if at all? For example, what does a relevance score of 0.4 mean, and how does it compare to one of 0.3? Do relevance scores across input features sum to the overall prediction probability? A brief description would be beneficial, as currently the assessment of important features appears to be in relation to other features from the same input.	We do not think that the absolute values of relevance scores can be interpreted in isolation. A patient with very little registered data can still have a high probability risk score, and thus get some high relevance scores from the layer wise relevance propagation analysis. So, relevance scores depend on how many features that are registered (not missing) that have been registered for the individual patient, what numeric values they have and the interaction between them? In relation to the question about if relevance scores across input features sum to the overall prediction probability? They sum to the logit, just before the last sigmoid transformation.	We have added the following sentence to the "Explanation module" section: "DTD obeys the conservation property describes by Montavan et al.²⁵ that ensures that the total redistributed relevance corresponds to the extent to which the illness of interest is detected by the function $f_c(\mathbf{x})$. In our case DTD is applied to the logits that feed into the final transformation layer."
1.16	- It is nice to see the authors list hyperparameters, however I could not find a crucial one: TCN kernel size/width. This should be listed as well.	We fully agree with the reviewer and refer to our answer to section 1.9.	We fully agree with the reviewer and refer to our answer to section 1.9.
1.17	- I take slight issue with the authors definition of a TCN, with one of the two hallmarks being that "TCNs can take a sequence of any length as input and output a sequence of the same length". I would argue that while this is indeed how TCNs can work, it is not a defining characteristic (since one may also desire this property in "standard" CNNs, without requiring causality) and is more related to the causal aspect of definition #1.	The reviewer has a valid point. We have changed the wording slightly, such that only the causal aspect is mentioned as a defining characteristic. The text "TCNs can take a sequence of any length as input and output a sequence of the same length" is still mentioned, but not as a defining characteristic.	The sentence not reads: "Moreover, TCNs differ from "ordinary" CNNs by at least one property: the convolutions in TCNs are causal in the sense that a convolution filter at time t is only dependent on the inputs that are no later than t, wherein the input subsequence is x_1, x_2, \dots, x_t. TCNs can take a sequence of any length as input and output a sequence of the same length, similar to RNNs^{33,39}."
Reviewer 2			
2.1	This paper reports on the application of an explainable ai method (namely Layer-wise relevance propagation (LRP)) on a neural network, which was conceived to perform a prediction from temporal data. The		No changes made in the document.

	data, preprocessing steps, neural network architecture, as well as the results of the relevance of each feature, are presented.		
2.2	The novel aspect is the domain that is used; because the medical domain is a very important domain. Other than that, there are a lot of aspects that the paper should have reasoned about; but the application domain justifies originality	We are glad that the reviewer finds our work suitable for publication in Nature Communications.	No changes made in the document.
2.3	Very important is that the authors mention (maybe in a sentence after line 42, page 1, or at the end) that explainability as it is described here is a technical precondition, but for a complete "explanatory loop" useful for the practical (!) medical domain expert the aspect of measuring how good an explanation fits to a certain problem and a certain domain expert (to whom it may be explained); consequently the authors should mention the concept of causability [xx] and how to measure causability [yy]: [xx] Holzinger, A., Langs, G., Denk, H., Zatloukal, K. & Mueller, H. 2019. Causability and Explainability of Artificial Intelligence in Medicine. Wiley Interdisciplinary Reviews: Data Mining and Knowledge Discovery, 9, (4), (2019) doi:10.1002/widm.1312. [yy] Holzinger, A., Carrington, A. & Müller, H. 2020. Measuring the Quality of Explanations: The System Causability Scale (SCS). Comparing Human and Machine Explanations. Springer/Nature KI - Künstliche Intelligenz (German Journal of Artificial intelligence), Special Issue on Interactive Machine Learning, Edited by Kristian Kersting, TU Darmstadt, 34, (2), doi:10.1007/s13218-020-00636-z.	We certainly agree with the reviewer; without a sufficient level of 'causability' there will be no practical benefit of an explanation. The intention of the sentence 'Clinicians must be able to understand the underlying reasoning of AI models so they can trust the predictions...' was to cover both aspects of what the paper [xx] terms as 1) 'explainability' (decision-relevant parts contributing to the prediction in the used representation for the AI model – in our case, the relevant input features), and 2) 'causability' (roughly, the degree to which the explanation interface conveys the machine statement into an explanation that supports the clinicians causal understanding – in our case, the simple visual representations of explanations). We have clarified this intend in a few sentences added to the second paragraph in our paper– as it was suggested by the reviewer. We have further added a reference to the 'System Causability Scale' from the paper [yy], as a measure to compare the quality of explanations from alternative methods in our discussion towards the end of the paper.	The following sentences were added to the end of the second paragraph to clarify what we mean by an understandable explanation: "Consequently, a useful explanation involves both the ability to account for the relevant parts in an AI model leading to a prediction, but also the ability to present this relevance in a way that supports the clinicians causal understanding in a comprehensible way [xx]. An explanation that is too hard to perceive and comprehend will most likely not have any practical effect." In the subsequent paragraph, we have also clarified that explanations are in the form of simple visual representations. Finally, we have added the following in our discussion towards the end of the paper: "An interesting subject for further study would also be to compare how well the explanation module in this paper conveys explanations to the clinical experts in various contexts compared to alternatives. To that end, Holzinger et al. recently proposed a Likert-scale based method tailored to explanations from AI."
2.4	The LRP method is briefly covered and in general the research work as it is described is correct. However, there are some important aspects missing, like e.g., perturbation analysis (masking tests). This method	We thank the reviewer for the suggestion about perturbation analysis, and confirm that it is a part of the innvestigate toolbox. While we agree that this could lead to other insights, we do not	We refer to section 1.3 for updates in the document.

	is contained in the innvestigate tool for the images, but should also be applied in this case, especially when the researchers state in page 2 that they wish to show robustness or in page 4 to show which parameters were the causes. This would give them much more insights than the (awaited) interpretation: The nearer to the disease time, the more relevant a feature is: “more weight to recent values” p.4. LPR papers go beyond that and show how the class prediction is affected by the gradual removal of the input features (pixels, words – word ablation) that are relevant and furthermore compare that to other eXAI methods. One important constraint that needs to be ensured for the application of this method is that the bias parameters are negative – this also can produce a negative effect in the performance of the NN. This is not mentioned and cannot be verified from the github repository as it is right now (this reviewer carefully checked this).	consider it crucial to add this analysis to the current results, which is already quite comprehensive for a study. As described in section 1.3 our model only has ReLU non-linearities, no additive bias in the linearities. The omission of bias terms did not seem to not affect model performance much, but we did not test this systematically.	
2.5	The results are clearly presented, but there is one exception: In p. 15 the baseline models are referred, but how are they related to the NN? LRP papers explain the dependency of a good explanation from the performance of the model.	We absolutely agree with the reviewer that explanation quality depends on model performance. Besides from that it is not entirely clear to the authors what is meant by this question. We apologize for this. If the reviewer would like to rephrase the question we will off course look into this.	No changes made in the document.
2.6	The paper is well written, easy to read and the figures are of good quality. Particularly p.6 is well written. Some problems occur in p.14 where the SAHPs are explained. It is not clear to the reviewer i.e. what the connection between the population-based LRP case has to do with Shapely values? A minor correction could be the use of the term “parameter” instead of “input feature” – “parameter” is usually used instead of neural network parameter.	We apologize for the confusion in relations to our use of the SHAP toolbox. We only used the plotting functions from the SHAP toolbox. We have added the sentence “The SHAP toolbox was not used to provide explanations” to make this clearer. In relations the use of the term “parameter” and “feature”. We agree that the word “parameter” could cause a misunderstanding as it refers to model parameters in technical papers. However, as this paper is also written for clinicians we chose not to use “feature” as	The sentence now reads: “The visual concepts of global parameter importance estimation and local explanation summary used in this paper are adopted from the SHapley Additive exPlanations (SHAP)⁵¹ library by Lundberg et al. The SHAP toolbox was not used to provide explanations.”

		word is not as common medical literature.	
--	--	---	--

Reviewers' Comments:

Reviewer #1:

Remarks to the Author:

The authors have addressed most of the initial comments. A few comments:

- In comment 1.4, I still think they should add something to the manuscript essentially summarizing their response.

- Regarding the cross-validation setup: it would be best if the authors could clarify their setup, whether it is a standard cross-validation + hold-out setup, or a nested cross-validation process.

Reviewer #2:

Remarks to the Author:

The authors have addressed all comments of the reviewers very adequately to the opinion of this reviewer, consequently this reviewer would now recommend accepting this paper.

Reviewer #3:

Remarks to the Author:

Nature communications

The authors developed an explainable artificial intelligence (AI) early warning score (xAI-EWS) system for early detection of acute critical illness. The system is based on the analysis of electronic health records (EHRs).

The approach has its limitations, but these are well discussed.

Comments

1-throughout the manuscript: ALI is obsolete: we abandoned it several years ago, to include ALI in ARDS (it is equivalent to mild ARDS).

2- Methods: organ dysfunction 'occurs when the SOFA score displays an acute increase of more than or equal to two points.': the SOFA score describes the degree of organ dysfunction, and any change must be taken into account. We selected the two points increase in the criteria for sepsis.

Minor : the proportion of men does not require two decimals.

Dear Reviewers

We thank you for your comments and feedback.

We have taken the liberty of dividing the comments into separate parts. Each part is seen in the table below with our comments and changes in the revised manuscript.

Best Regards
The authors

Number	Reviewer comments	Author comments	Change in revised manuscript
Reviewer 1			
	The authors have addressed most of the initial comments. A few comments: - In comment 1.4, I still think they should add something to the manuscript essentially summarizing their response.	We agree with the reviewer, and have added a paragraph about this in the discussion.	The following text have been added to the discussion: “The development of the models in this study was done in an iterative way where results from technical development were continuously presented to, and discussed with, clinicians from an emergency department. The primary purpose of this iteration process was to ensure that the models learned at least some correlations that are already considered established knowledge in the clinical field. It would be obvious to try to use this technology hypothesis-generating, whereby output from LRP analysis is used as inspiration to discover new and unknown correlations.”
	- Regarding the cross-validation setup: it would be best if the authors could clarify their setup, whether it is a standard cross-validation + hold-out setup, or a nested cross-validation process.	We agree with the reviewer, and have modified the model evaluation paragraph. We also added a new figure (Supplementary Figure 1) to the supplementary information illustrating the cross validation scheme.	The following text have been added to the discussion: “The xAI-EWS model was validated using five-fold cross-validation. Data were randomly divided into 5 portions of 20% each. For each fold four portions (80 %) was used to fit the xAI-EWS model parameters during training. The remaining 20% was split into two portions of 10% each for validation and test. The validation data were used to perform an unbiased evaluation of a model fit during training, and the test data were used to provide an unbiased evaluation of the final model. All data for a single patient was assigned to either train, validation or test data. Figure 2

			report performance from the test data. For each fold data were shifted such that a new portion was used for testing. The cross validation scheme is illustrated in Supplementary Figure 1. As comparative measures, we used the area under the receiver operating characteristic curve (AUROC) and the area under the precision-recall curve (AUPRC).”
Reviewer 2			
	The authors have addressed all comments of the reviewers very adequately to the opinion of this reviewer, consequently this reviewer would now recommend accepting this paper.	We thank for the positive evaluation.	
Reviewer 3			
	The authors developed an explainable artificial intelligence (AI) early warning score (xAI-EWS) system for early detection of acute critical illness. The system is based on the analysis of electronic health records (EHRs). The approach has its limitations, but these are well discussed.	We acknowledge the point made by the reviewer. All retrospective data analysis by nature contain bias' - known or unknown. But we thank the reviewer for appreciating our discussion of these.	
	1-throughout the manuscript: ALI is obsolete: we abandoned it several years ago, to include ALI in ARDS (it is equivalent to mild ARDS).	We thank the reviewer for clarifying the relationship between ALI and ARDS. In this study we based the ground truth for ALI on the need for Continuous Positive Airway Pressure (CPAP) or Noninvasive ventilation (NIV) because PaO₂/FiO₂ measurements were not available. This have now been written explicitly in the manuscript.	The methods sections now reads: “For ALI classification, we considered the presence of either NIV or CPAP during the admission, because PaO₂/FiO₂ measurements were not available. The ALI onset was the first occurrence of either NIV or CPAP (see Figure 5).” The discussion section now reads: “We based the ground truth on the need for Continuous Positive Airway Pressure (CPAP) or Noninvasive ventilation (NIV) because PaO₂/FiO₂ measurements were not available.”
	2- Methods: organ dysfunction ‘occurs when the SOFA score displays an acute increase of more than or equal to two points.’: the SOFA score describes the degree of organ dysfunction, and any change must be taken into account. We selected the two points increase in the criteria for	The reviewer has a valid point. We have updated the statement accordingly.	The text now reads: “The degree of organ dysfunction is described by an acute increase in the SOFA score and an increase of more than or equal to two points is used in the criteria for sepsis.”

	sepsis.		
	Minor: the proportion of men does not require two decimals.	Updated to one decimal	Updated to one decimal